# REPRESENTATION LEARNING THROUGH LATENT CANONICALIZATIONS

## ABSTRACT

We seek to learn a representation on a large annotated data source that generalizes to a target domain using limited new supervision. Many prior approaches to this problem have focused on learning "disentangled" representations so that as individual factors vary in a new domain, only a portion of the representation need be updated. In this work, we seek the generalization power of disentangled representations, but relax the requirement of explicit latent disentanglement and instead encourage linearity of individual factors of variation by requiring them to be manipulable by learned linear transformations. We dub these transformations **latent canonicalizers**, as they aim to modify the value of a factor to a pre-determined (but arbitrary) canonical value (e.g., recoloring the image foreground to black). Assuming a source domain with access to meta-labels specifying the factors of variation within an image, we demonstrate experimentally that our method helps reduce the number of observations needed to generalize to a similar target domain when compared to a number of supervised baselines.

## 1 INTRODUCTION

Most state-of-the-art visual recognition models rely on supervised learning using a large set of manually annotated data (Krizhevsky et al., 2012; He et al., 2015; He et al., 2017; Redmon & Farhadi, 2018). As recognition task complexity increases, so does the number of potential real world variations in visual appearance and hence the size of the example set needed for sufficient test time generalization. Unfortunately, large labeled data sets are expensive to acquire (Deng et al., 2009; Zhou et al., 2017), and may even be infeasible for applications with evolving data distributions.

Often a large portion of the variance within a collection of data is due to task-agnostic factors of variation. For example, the appearance of a street scene will change substantially based on the time of day, weather pattern, and number of traffic lanes, regardless of whether cars or pedestrians are present. Ideally, the ability to recognize cars and pedestrians would not require labeled examples of street scenes for all combinations of times of day, weather conditions, and geographic locations. Rather it should be sufficient to observe examples from each factor independently and generalize to unseen combinations. However, often the in-domain labeled data available may not even linearly cover all factors of variation. This calls for methods that encourage such sample efficiency by focusing on the individual complexities of the factors of variation, as opposed to their product.

Prior approaches to learning representations which isolate factors of variation in the data have typically regularized the representation itself, with the aim of learning "disentangled" representations (Kulkarni et al., 2015; Chen et al., 2016; Higgins et al., 2017a;b; Burgess et al., 2018; Kim & Mnih, 2018). Alternatively, one might regularize the *way the representation can be manipulated* rather than the representational structure itself. Here, we take such an approach by introducing **latent canonicalization**, in which we constrain the representation such that individual factors can be clamped to an arbitrary, but fixed value ("canonicalized") by a simple linear transformation of the representation. These canonicalizers can be applied independently or composed together to canonicalize multiple factors.

We assume access to a large collection of source domain data with meta-labels specifying the factors of variation within an image. This may be available from meta-data, attribute labels, or from hyperparameters used for generation of simulated imagery. Latent canonicalizers are optimized by a pixel loss over pairs of ground-truth canonicalized examples and reconstructions of images with various

factors of variation whose representation has been passed through the relevant latent canonicalizers. By requiring the ability for manipulation of the latent space according to factors of variation, latent canonicalization encourages our representation to linearize such factors.

We evaluate our approach on its ability to learn general representations after observing only a subset of all potential combinations of factors of variation. We first consider the simple dSprites dataset, introduced to study disentangled representations (Matthey et al., 2017) and show qualitatively that we can effectively canonicalize individual factors of variation. We next consider the more realistic, though still tractable, task of digit recognition on street view house numbers (SVHN) (Netzer et al., 2011) with few labeled examples. Using a digit simulator, we train our representation with latent canonicalization along multiple axes of variation such as font type, background color, etc. and then use the factored representation to enable more efficient few-shot training of digit recognition in the real data. Our method substantially increased performance over standard supervised learning and fine-tuning for equivalent amounts of data, achieving digit recognition performance that was only attainable with $5\times$ as much SVHN labeled training data for the best-performing baseline method. Our experiments offer promising evidence that encoding structure into the latent representation guided by known factors of variation within data can enable more data efficient learning solutions.

## 2 RELATED WORK

### 2.1 DISENTANGLING

A number of studies have sought to learn low-dimensional representations of the world, with many aiming to learn "disentangled" representations. In disentangled representations, single latent units independently represent semantically-meaningful factors of variation in the world and can lead to better performance on some tasks (van Steenkiste et al., 2019). This problem has been most commonly studied in an unsupervised setting, often by regularizing latent representations to stay close to an isotropic Gaussian (Higgins et al., 2017a;b; Burgess et al., 2018; Kim & Mnih, 2018). Other popular unsupervised approaches include maximizing the mutual information between the latents and the observations (Chen et al., 2016) and adversarial approaches (Donahue et al., 2016; Radford et al., 2015). When supervision on the sources of variation is available, it is possible to use this in a weak way (Kulkarni et al., 2015).

Many of these works have explicitly endeavored to learn semantically meaningful representations which are both linearly independent and axis-aligned, such that individual latents correspond to individual factors of variation in the world. However, recent work has questioned the importance of axis-aligned representations, showing that many of these methods rely on implicit supervision and finding little correspondence between this strict definition of disentanglement and learning of downstream tasks (Rolinek et al., 2018; Locatello et al., 2019). Further, while axis-alignment is useful for human interpretability, for the purposes of decodability, any arbitrary rotation of these latents would be equally decodable so long as factors are linearly independent (Locatello et al., 2019). In this work, we use explicit supervision in a simulated setting to encourage linear, but not necessarily axis-aligned representations.

### 2.2 SIM2REAL

The setting of near-unlimited simulated data with ground truth labels and scarce real data occurs often in computer vision and robotics. However, the *domain gap* between simulated and real data reduces generalization capacity. Many approaches have been proposed to overcome this difficulty which are broadly referred to as *sim2real* approaches. A simple approach to closing the sim2real gap is to train networks with combinations of real and synthetic data (Varol et al., 2017).

**Transfer Learning and Few-shot Learning:** Alternatively, synthetic data can be used for pre-training followed by fine-tuning on real data (Richter et al., 2016)—a form of transfer learning. Often, the reason one uses simulated data is that there is a shortage of labeled real data. In this situation, one may make use of few-shot learning techniques which seek to prevent over-fitting to the few examples by using a metric loss between data tripets (Koch et al., 2015), comparing similarity between individual examples (Vinyals et al., 2016) or between a prototypical example per category and each instance (Snell et al., 2017). A separate class of techniques use meta-learning to design a learning solution which is most amenable to learning from few examples (Finn et al., 2017). However, these approaches may be sensitive to the ratio of synthetic and real data.

**Domain adaptation:** With access to a large set of unlabeled real examples, domain adaptation techniques can be used to close the sim2real gap. One class of techniques focuses on matching domains at the feature level (Ganin & Lempitsky, 2015; Long et al., 2015; Tzeng et al., 2017), aiming to learn domain-invariant features than can make models more transferrable (Zou et al., 2018; Li et al., 2018a). In fact, image-to-image translation focuses on the appearance gap by bridging the *appearance gap* in the image domain instead of feature space (Shrivastava et al., 2017; Liu et al., 2017). Domain adaptation can also be used to learn *content gap* in addition to appearance gap (Kar et al., 2019). Additional structural constraints, such as cycle consistency, can further refine this image domain translation (Zhu et al., 2017; Hoffman et al., 2018; Mueller et al., 2018). Finally, image stylization methods can also be adapted for style transfer and sim2real adaptation (Li et al., 2018b).

**Domain randomization:** (DR) offers a surprising alternative (Tobin et al., 2017; Prakash et al., 2018) by exploiting control of the synthetic data generation pipeline to randomize sources of variation. Random variations will likely be ignored by networks and thus result in invariance to those variations. A particularly interesting instantiation of DR was suggested in Sundermeyer et al. (2018) for pose estimation. Pose is an example of a factor of variation which could be ambiguous due to occlusions and symmetries. Instead of explicitly regressing for the pose angle, the authors propose an implicit latent representation. This is achieved by an augmented-autoencoder, a form of denoising-autoencoder that addresses all nuisance factors of variation as noise. This idea can be seen as a particular case of our method where all factors are being canonicalized at once instead of learning to canonicalize each individually. Another interesting example is the quotient space approach of (Mehr et al., 2018) removes pose information for a 3D shape representation by max-pooling encoder features over a sampling of object rotations. It, however, does not consider how to perform canonicalization as a linear transformation in latent space, nor how to compose different canonicalizers.

## 3 APPROACH

Our goal is to learn representations which capture the generative structure of a dataset by independently representing the underlying factors of variation present in the data. While many previous works have approached this problem by regularizing the representation itself (Kulkarni et al., 2015; Chen et al., 2016; Higgins et al., 2017a;b; Burgess et al., 2018; Kim & Mnih, 2018), here we take a different approach: rather than directly encourage the representation to be disentangled, we instead encourage the representation to be structured such that individual factors of variation can be manipulated by a simple linear transformation. In other words, we constrain the *way that the representation can be manipulated* rather than the *structure of the representation itself*.

### 3.1 LATENT CANONICALIZATION

In our approach, we augment a standard convolutional denoising autoencoder (AE) with **latent canonicalizers**. A standard AE learns an encoder, Enc, which takes as input a given image, $x$, and produces a corresponding latent vector, $z$. At the same time, the latent vector is used as input to a decoder, Dec, which produces an output image, $\hat{x}$. Both the encoder and decoder are learned according to the following objective, $\mathcal{L}_{\text{ae}}$, which minimizes the difference between the original input image and the reconstructed output image:

$$z = \text{Enc}(x; \theta_e) \quad ; \quad \hat{x} = \text{Dec}(z; \theta_d) ; \tag{1}$$

$$\mathcal{L}_{\text{ae}}(\theta_e, \theta_d; x) = \min_{\theta_e, \theta_d} \|x - \hat{x}\|_2^2. \tag{2}$$

To encourage noise-robustness, we augment the potential input images following previous work on denoising autoencoders (Vincent et al., 2008; 2010), noising each raw input image, $x$, by adding Gaussian noise, blur, and random crops and rotations, leading to our noised input image, $\tilde{x}$.

In this work, we additionally constrain the structure of the learned latent space using a set of latent canonicalization losses. We define a latent canonicalizer as a learned linear transformation, $\mathcal{C}$, which operates on the latent representation, $z$, in order to transform a given factor of variation (e.g., color or scale) from its original value to an arbitrary, but fixed, canonical value. So that individual factors can be manipulated separately, we learn unique canonicalization matrices, $\mathcal{C}_j$, for each factor of

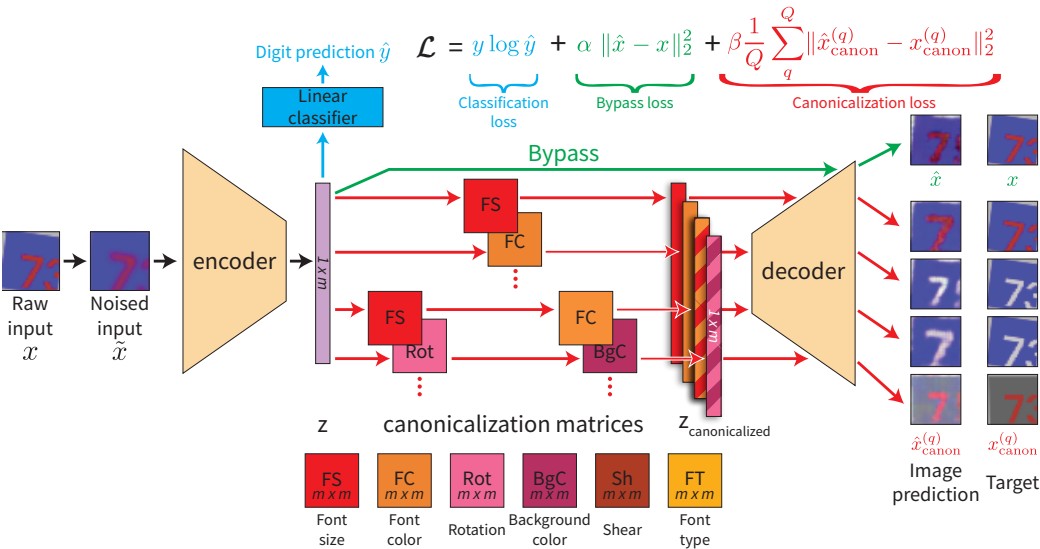

**Figure 1: Schematic representation of latent canonicalization:** Colored paths correspond to different components of the loss (cyan: classification, green: bypass, red: canonicalization). Four possible canonicalizations (two individual and two pairs) are shown along with example simulated SVHN images and reconstructions.

variation, $j \in [1, K]$, present in the dataset. In order to constrain the latent representation according to canonicalization along one factor, $j$, our method yields the following basic form:

$$z_{\text{canon}}^{(j)} = \text{Enc}(\tilde{x}; \theta_e) \cdot \mathcal{C}_j \quad ; \quad \hat{x}_{\text{canon}}^{(j)} = \text{Dec}(z_{\text{canon}}; \theta_d) \tag{3}$$

To supervise the learning of latent canonicalizers, we compare the images generated by canonicalized latents, $\hat{x}_{\text{canon}}$, to ground truth images with the appropriate factors of variation set to their canonical values, $x_{\text{canon}}$. Canonicalizers can also be composed together to canonicalize multiple factors (e.g., $z_{\text{canon}}^{(j,k)} = z \cdot \mathcal{C}_j \cdot \mathcal{C}_k$). During training, each image is passed through a random subset of both individual and pairs of canonicalizers. Given $Q$ canonicalization paths for a given image, $x$, the corresponding canonicalization loss for that single image (red in Figure 1) is written as:

$$\mathcal{L}_{\text{canon}} = \frac{1}{Q} \sum_q^Q \|\hat{x}_{\text{canon}}^{(q)} - x_{\text{canon}}^{(q)}\|_2^2 \tag{4}$$

Since many outputs are canonicalized, it is possible that the decoder will simply learn to only generate the canonical value of a given factor of variation. To prevent this form of input-independent memorization, we also include a "bypass" loss which is equivalent to the standard denoising auto-encoder formulation (green in Figure 1) defined in Equation (2), thus forcing information about each factor to be captured in the latent vector, $z$.

Finally, we ensure that our representation not only allows for linear manipulation along factors of variation, but does so while capturing the information necessary to train a classifier to solve our end task. To this end, we add a supervised cross-entropy loss, $\mathcal{L}_{\text{CE}}$, which optimizes our end task using available labeled data (cyan in Figure 1):

$$\mathcal{L}_{\text{CE}} = y \log \hat{y} \tag{5}$$

Combining equations 2, 4, and 5 with loss-scaling factors ($\alpha$ and $\beta$) gives us our final per-example loss formulation:

$$\mathcal{L} = y \log \hat{y} + \alpha \|\hat{x} - x\|_2^2 + \beta \frac{1}{Q} \sum_q^Q \|\hat{x}_{\text{canon}}^{(q)} - x_{\text{canon}}^{(q)}\|_2^2 \tag{6}$$

In practice, two canonicalizers are chosen at random for each input batch, $\mathcal{C}_h$ and $\mathcal{C}_j$, and the corresponding latent representation $z$ is passed through $\{\mathcal{C}_h, \mathcal{C}_j, \mathcal{C}_h\mathcal{C}_j, \mathcal{C}_j\mathcal{C}_h\}$ generating four unique canonicalized latents: $\{z_{\text{canon}}^{(h)}, z_{\text{canon}}^{(j)}, z_{\text{canon}}^{(h,j)}, z_{\text{canon}}^{(j,h)}\}$. A diagram of the method is shown in Figure 1, illustrating each canonicalization path, the bypass path, and the classification model. All description thus far has focused on the single image loss for simplicity of presentation. The full model averages the per image loss over mini-batch before making gradient updates.

**Latent canonicalizer constraints:**   Our approach relies upon constraining the way a representation can be manipulated. As a result, the specific choice of constraints should have a significant impact on the representations which are ultimately learned. Here, we limit ourselves to only two constraints: the transformations must be linear and canonicalizers must be composable, at least in pairs. If we were to allow non-linear canonicalizers, there would be little incentive for the encoder to learn an easily manipulable embedding. This would only be exacerbated as the non-linear canonicalizer is made more powerful by e.g., additional depth. By requiring canonicalizers to be composable, we encourage independence as each canonicalizer must be able to be applied without damaging any other. We explore some other potential constraints in Section 4.2.4.

## 3.2   SIM2REAL EVALUATION

A main motivation of latent canonicalization is to leverage structure gleaned from a large source of data with rich annotations to better adapt to downstream tasks. A natural such setting for performance evaluation is sim-to-real. Specifically, we make use of the Street View House Numbers (SVHN) dataset as our real domain. To simulate SVHN, we built a SVHN simulator in which we have full control over many factors of variation such as font, colors and geometric transformations (a detailed description of the simulator is given in Section 4.2.1).

We first pre-train on the synthetic data with latent canonicalization. Following this step, we freeze the canonicalizers and investigate whether the learned representations can be leveraged as the input to a linear classifier for few-shot learning on real examples, labeled only with the class of interest (e.g., no meta-labels for additional factors of variation). During this stage, the encoder is also refined.

**Majority vote:**   Because latent canonicalization manipulates the latent representation, we can use canonicalization as a form of "latent augmentation." In this setting, we can aggregate the predictions of the digit classifier across many canonicalization paths, each of which confers a single "vote." Critically, such an approach requires the ability to cleanly manipulate the learned representation, and is therefore only possible for our proposed method. For a more detailed exploration of the impact of majority vote, see Section 4.2.4.

## 3.3   BASELINES

We compare our proposed latent canonicalization with several baselines. To ensure the most fair comparison possible, we aimed to fix as many hyperparameters as possible: we use the same backbone architecture in all our network modules (a detailed description of our network is given in Section 3.1); the same number of epochs at the pre-training stage (learning from synthetic data); and a carefully chosen number of epochs at the refinement stage to fit well the method overfitting rate. For all latent canonicalization experiments we trained three models on the simulated data and then performed five refinements of each pre-trained model, for a total of fifteen replicates. Results are reported as mean ± standard deviation. For all baseline models, 15 independent replicates were trained.

Our simplest baseline, which is meant as a lower bound, is simply a classifier trained only on the low-shot real data. Next is a classifier trained on synthetic data and then refined on low-shot real data. The vanilla-AE is an autoencoder pre-trained on synthetic data, after which a linear classifier is trained on low-shot real data, allowing the encoder to be refined. The strongest baseline is an autoencoder augmented with a digit classifier at both pre-training (on simulated data) and at refinement. The loss weighting was chosen via a hyper-parameter search individually for each model.

## 4   EXPERIMENTS

### 4.1   LATENT CANONICALIZATION OF DSPRITES

Key to our method is the use of latent canonicalizers, which are learned linear transformations optimized to eliminate individual factors of variation. As a first test of the effectiveness of latent canonicalization, we evaluated our framework using the toy data set, dSprites (Matthey et al., 2017), which was designed for the exploration and evaluation of disentanglement methods. Specifically, dSprites is a dataset of images of white shapes on a black background generated by five independent factors of variation (shape, scale, rotation, and x- and y-positions). Training our model (Figure 1) on dSprites, we demonstrate the effect of applying different individual canonicalizers to various values

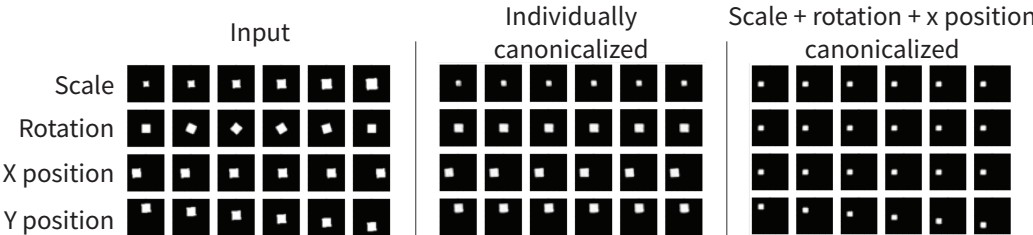

**Figure 2: Canonicalization of dsprites images:** Input dsprites images (left), reconstructions of inputs with one factor canonicalized (middle), and rotation, scale, and x-position canonicalized (right). Each row demonstrates how images change as a single factor of variation is altered.

of the input factors (Figure 2 left and middle). We therefore also applied a set of three canonicalizers (scale, rotation, and x-position) sequentially as shown in Figure 2, right. Encouragingly, we found that not only did individual canonicalizers effectively canonicalize their factor of interest, multiple canonicalizers can be applied in sequence. Furthermore, although models were trained with only pairs of canonicalizers, triplets of canonicalizers also performed well (Figure 2, right).

## 4.2 LATENT CANONICALIZATION OF SVHN

To evaluate the impact of latent canonicalization in a more realistic, though still controllable setting, we turned to the well-known Street View House Numbers (SVHN) dataset (Netzer et al., 2011). In Section 4.2.1, we discuss our approach to designing a simulator for SVHN, which we use in Section 4.2.2 to pre-train our models and measure the impact of latent canonicalization on sim2real transfer, finding that latent canonicalization substantially improved performance relative to our strongest baseline. In Section 4.2.3, we investigate the structure of the learned representations to measure the representational impact of latent canonicalization. Finally, in Section 4.2.4, we discuss potential directions for improvement which, unfortunately, either harmed or left unchanged sim2real performance relative to our best-performing models.

### 4.2.1 SIMULATING SVHN

To support our proposed training procedure, we require a comprehensive dataset with detailed metadata regarding ground-truth factors of variation. While this is possible for a natural dataset, such data can also be generated for visually realistic, but fairly simplistic datasets such as SVHN. To this end, we built a procedural image generator that simulates the SVHN dataset by rendering images with digits on a constant-colored background (see examples in Figure A1). Apart from the digit class variation, we also simulate additional factors of variation: font color, background color, font size, font type, rotation, shear, fill color for newly created pixels, scale, number of digit instances, translation, Gaussian noise, and blur. A detailed description of the simulator is provided in Section C. Among these factors we chose the first six as our supervised factors of variation, noise and blur as a joint noise model, and the rest as additional factors which enrich the variety of the data without supervised canonicalization. Some of the resulting images can be seen in Figure 3[1]. To enable reproducability across comparisons, and to minimize unaccounted for variability in the data, we generated a fixed training set with 75,000 images along with targets for all possible canonicalization paths. We emphasize that this training set represents a small fraction ($\sim 0.2\%$) of the total number of possible combinations of factors. We used such a small fraction of the total space to demonstrate that latent canonicalization is feasible even if the factor space is only sparsely sampled. The simulator along with the generated train set will be made publicly available.

### 4.2.2 SIM2REAL SVHN TRANSFER USING LATENT CANONICALIZATION

We want to learn representations which enable models to generalize to novel data with consistent underlying structure. Moreover, the effectiveness of disentangling for acquisition of downstream tasks has recently been called into question (Locatello et al., 2019). We therefore evaluate the quality of our learned representations by measuring their ability to adapt to real examples. Specifically, we

---

[1]see Appendix C for further simulator details

| Model | 10 shot | 20 shot | 50 shot | 100 shot | 1000 shot |
|---|---|---|---|---|---|
| **Classifier trained on real only** | $20.86 \pm 2.03$ | $36.08 \pm 2.49$ | $72.75 \pm 1.99$ | $82.24 \pm 0.78$ | $\mathbf{92.84 \pm 0.39}$ |
| **Classifier trained on synth** | $76.50 \pm 2.06$ | $80.07 \pm 1.09$ | $83.95 \pm 0.79$ | $85.65 \pm 1.17$ | $89.82 \pm 0.60$ |
| **Vanilla AE** | $15.39 \pm 6.12$ | $21.55 \pm 9.16$ | $41.98 \pm 16.62$ | $51.18 \pm 19.80$ | $58.51 \pm 19.68$ |
| **AE + Classifier** | $79.61 \pm 1.18$ | $81.63 \pm 1.00$ | $84.66 \pm 0.76$ | $85.86 \pm 0.82$ | $88.80 \pm 0.62$ |
| **Ours** | $82.55 \pm 0.86$ | $84.83 \pm 0.76$ | $87.82 \pm 0.57$ | $\mathbf{89.40 \pm 0.48}$ | $91.21 \pm 0.24$ |
| **Ours + majority vote** | $\mathbf{83.41 \pm 1.23}$ | $\mathbf{85.41 \pm 0.88}$ | $\mathbf{88.17 \pm 0.53}$ | $\mathbf{89.58 \pm 0.57}$ | $91.34 \pm 0.34$ |

**Table 1: sim2real transfer results** Model performance on the SVHN test set using low-shot labeled real examples for method and baselines. Table entries represent mean ± std.

consider a few-shot setting in which we allow models pre-trained on simulated data access to a small refinement set of a few annotated examples per class. We ran this experiment with per-class set sizes of 10, 20, 50, 100 and 1000. To measure sim2real transfer, we train a fresh linear classifier on the representation learned by the encoder pre-canonicalization, $z$ , while allowing the encoder to be refined as well. We report accuracy on the unseen SVHN test set. As a measure of the pre-trained model, we include examples of reconstructions generated by canonicalized latents in Figure 3.

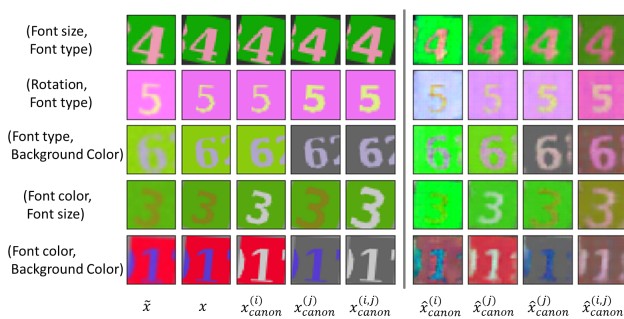

$\tilde{x}$  $x$  $x_{canon}^{(i)}$  $x_{canon}^{(j)}$  $x_{canon}^{(i,j)}$  $\hat{x}_{canon}^{(i)}$  $\hat{x}_{canon}^{(j)}$  $\hat{x}_{canon}^{(j)}$  $\hat{x}_{canon}^{(i,j)}$

**Figure 3: Example targets and reconstructions of canonicalized simulated SVHN images.**

When small train-sets are used, results may vary substantially depending on the selected set. To account for this, we (a) use the same train set for all methods, and (b) ran each experiment 15 times: 3 different networks were trained on simulated data with different random seeds and 5 replicate refinements were performed per pre-trained network.

Table 1 shows the sim2real SVHN results for our method along with four baselines: a classifier trained solely on low-shot real data, a classifier trained on a large dataset of synthetic data and refined on low-shot real data, a linear classifier trained on the bottleneck of a vanilla denoising autoencoder trained on synthetic data, and a linear classifier trained on an autoencoder pre-trained with a digit classifier on synthetic data. For all settings with fewer than one thousand examples per class, we found that our model outperformed the best competing baseline by $\sim 3-4\%$. We further improved our model's performance by taking a "majority vote" approach to inference type classification, in which we pass the representation through multiple latent canonicalizers in parallel to generate multiple votes as discussed in Section 3.2. Consistently, we found that majority vote boosted performance, by up to $\sim 0.9\%$, with the largest gains coming for the lowest-shot settings.

To contextualize the importance of this improvement, one can see that to match our reported performance on 20 shot with the best baseline of an AE + Classifier, a 5 times larger train set of 100 is required. This demonstrates the potential of our proposed method in better utilizing access to meta-labels for better adaptation to real data.

### 4.2.3 ANALYSIS OF REPRESENTATIONS

**Linear decodability of factors of variation from representations:** While latent canonicalization encourages representations to be linearly manipulable, it does not explicitly encourage linear decodability. However, since our canonicalizers are constrained to be linear, we conjecture that latent canonicalization may also encourage linear decodability. To test this quantitatively, we trained linear classifiers on the pre-trained, frozen encoder for each factor of variation. We ran this experiment separately for each factor and compared linear decodability to our best baseline, AE+Cls. For continuous factors of variation, we binned target values, converting the problem into a multi-class classification task. Importantly, the number of bins differs across factors, so chance performance is different per factor. For background color, font color, and rotation angle, the canonicalized representation was noticeably more linear than the baseline (shown here by higher accuracy on a held-out

Rotation

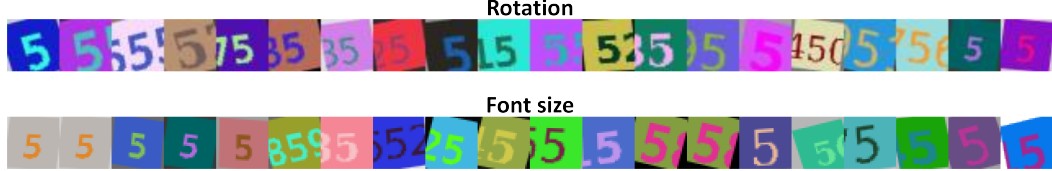

Font size

**Figure 5: Visualization of the linear properties of the representation learned by the canonicalizers.** Each row shows the first principal component of $z_{\text{canon}}^{(j)} - z$ for a source of variation. A clear pattern is visible for rotation (left to right tilted), and font size (small to big). 20 normally distributed samples from a batch of 1000 are shown above. See appendix for a visualization of the principal components of the bypassed latent.

test set), whereas font type showed a smaller improvement and font size and shear showed no improvement in linear decodability (Figure 4). One possible explanation for the discrepancy across factors is that font color, background color and, rotation are the most visually salient factors with the largest range of variability. We leave a more thorough analysis of the precise features which make factors linearly decodable to future work.

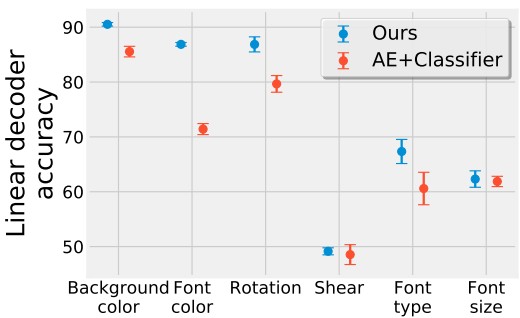

**Figure 4: Linear decodability of factors of variation.** Performance of a linear decoder trained on the frozen, pre-canonicalized representation, $z$ for each of the six factors of variation. Each factor was divided into a different number of bins for multi-class classification, such that chance is 1.6% for background and font color, 33.3% for rotation, 10% for shear, and 16.7% for font type and size. Error bars represent mean $\pm$ std across three pre-trained networks.

**Visualizing the impact of canonicalization:** In the previous experiment, we quantified the linear decoding performance of our representations; here, we attempt to visualize the extent to which they are linear. If these representations were indeed linear, we would expect them to be easily decomposable using principal component analysis (PCA), the components of which we can visualize. However, the latent codes from each of the canonicalizers *removes* the effect of a source of variation while keeping the others. We therefore compute the principal components (PCs) of $z_{\text{canon}}^{(j)} - z$, i.e., the difference between a canonicalized latent and the pre-canonicalized latent, such that PCs now represent the *removed* factor of variation. In Figure 5, we show sorted images along the first PC, showing a clear linear sorting of rotation, and font size. This visually demonstrates that our approach is able to extract latents that have strong linearity.

### 4.2.4 DISCUSSION OF ALTERNATIVE DESIGN DECISIONS

Latent canonicalization opens up many additional avenues for modification to potentially produce better representations and, consequently, better sim2real performance. In the previous section, we showed how incorporating majority vote further increased performance. Here, we discuss several other modifications we explored, which resulted in no change or a decrease in sim2real performance.

**Idempotency reconstruction loss:** To encourage composability of canonicalizers, we trained them in identical pairs with alternating orders for consistency (e.g., $z \cdot \mathcal{C}_1 \mathcal{C}_2$ and $z \cdot \mathcal{C}_2 \mathcal{C}_1$). We also tried encouraging idempotency, such that the same canonicalizer can be repeatedly applied without changing the reconstruction (e.g., $z \cdot \mathcal{C}_1 \mathcal{C}_2$). We found that applying this loss actually *harmed* performance, reducing pre-majority vote classification accuracy by $\sim 0.5\%$ (Table A1, second row).

**Classifier location during pre-training:** In our model, the classifier is placed at the output of the encoder prior to canonicalization, $z$. However, one might imagine that latent canonicalization could serve as a form of data augmentation, such that placing the classifier after the canonicalization step, $z_{\text{canon}}$Canon, might increase performance. In contrast, we found that placing the classifier after the canonicalization step harmed performance, reducing pre-majority vote classification accuracy by $\sim 3\%$ (Table A1, third row). Interestingly, however, we found that the majority vote method, which can also be viewed as a form of data augmentation, did in fact increase performance.

**Latent consistency and idempotency loss:** The impact of latent canonicalization was supervised at the image level, by comparing the reconstruction to a target image. However, we could also use a self-supervised loss at the latent level, by enforcing consistency (i.e., $\min\|z \cdot C_1 C_2 - z \cdot C_2 C_1\|_2$) and idempotency (i.e., $\min\|z \cdot C_1 C_1 - z \cdot C_1\|_2$). To account for the scale of this latent loss, which was much larger than the other loss components, we used a very small scale factor for the latent loss to maximize performance (1e-7). Even with an appropriately scaled loss, however, the latent loss either had little impact or harmed sim2real performance (Table A1, fourth row).

**Alternative majority votes:** We found that a simple majority vote containing the pre-canonicalized and individually canonicalized representations (1 and 6 votes, respectively) increased performance by $\sim 0.5\%$. To further augment the vote set, we also tried adding votes via idempotency (6 additional votes) and pairs (30 additional votes). We found that neither of these additions further improved performance over the simplest majority vote approach (Table A2).

## 5  CONCLUSION

We have introduced the notion of **latent canonicalization**, in which we train models to manipulate latent representations through constrained transformations which set individual factors of variation to fixed values ("canonicalizers"). We demonstrate that latent canonicalization encourages representations which result in markedly better sim2real transfer on the SVHN dataset than comparable models, even when only a small sample of the possible combination space was used for training. Notably, latent canonicalized pre-trained models reached few-shot performance which required $5\times$ as much data for comparable baselines. Analyzing the representations themselves, we found that the representation of factors of variation was linearized, as measured by decodability and linear dimensionality reduction (PCA). Finally, we discuss a number of alternative constraints which did not help performance. Here, we primarily analyzed SVHN, a realistic, but relatively simple dataset. However, the strong performance of latent canonicalization on SVHN is encouraging for larger-scale data with more complex factors of variation. Our results suggest the promise not only of latent canonicalization, but, more broadly, methods which encourage representational structure by constraining representational transformations rather than a particular structure itself.

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

# A  ALTERNATIVE DESIGN DECISIONS − ADDITIONAL MATERIAL

| Model | 10 shot | 20 shot | 50 shot | 100 shot | 1K shot |
|---|---|---|---|---|---|
| **Ours (no maj vote)** | $82.55 \pm 0.86$ | $84.83 \pm 0.76$ | $87.82 \pm 0.57$ | $89.40 \pm 0.48$ | $91.21 \pm 0.24$ |
| **Ours + idem** | $81.77 \pm 1.23$ | $84.08 \pm 0.98$ | $87.08 \pm 0.64$ | $88.96 \pm 0.51$ | $90.74 \pm 0.35$ |
| **Ours + classifier after** | $79.69 \pm 1.22$ | $81.14 \pm 1.00$ | $84.21 \pm 0.74$ | $86.42 \pm 0.54$ | $88.62 \pm 0.24$ |
| **Ours + latent loss 1e-7** | $82.74 \pm 0.58$ | $84.74 \pm 0.84$ | $87.47 \pm 0.50$ | $88.88 \pm 0.33$ | $90.58 \pm 0.18$ |

**Table A1: Summary of model additions which did not change or harmed performance.** Table entries represent mean ± std.

| Model | 10 shot | 20 shot | 50 shot | 100 shot | 1K shot |
|---|---|---|---|---|---|
| **Ours with maj vote** | $83.41 \pm 1.23$ | $85.41 \pm 0.88$ | $88.17 \pm 0.53$ | $89.58 \pm 0.57$ | $91.34 \pm 0.34$ |
| **Ours + majority vote idem** | $83.44 \pm 1.21$ | $85.44 \pm 0.87$ | $88.20 \pm 0.55$ | $89.60 \pm 0.58$ | $91.35 \pm 0.36$ |
| **Ours + majority vote all-pairs** | $83.26 \pm 1.12$ | $85.26 \pm 0.80$ | $88.14 \pm 0.53$ | $89.59 \pm 0.58$ | $91.33 \pm 0.33$ |

**Table A2: Summary of alternative majority vote methods.** Table entries represent mean ± std.

# B  NETWORK IMPLEMENTATION DETAILS

All the models in this paper are based on the modules: Encoder (Enc), Decoder (Dec), latent canonicalizers (Can) and a linear classifier (Cls). Specifically the classifier baseline is a composition of Enc+Cls; the AE is a composition Enc+Dec. For AE+Cls we compose a linear classifier and supervised for both reconstruction and classification loss. For our proposed network, we also have 6 latent canonicalizers, used both individually and in pairs.

**Loss weights:**  We performed a hyper-parameter sweep over the classification loss weight ($\in$ $10, 20, 50$ and found 50 to be the strongest baseline. We also used 50.0 as the classification loss for our network.

**Network architecture:**  The Encoder is comprised of 3 modules, each includes 3 layers of 3x3 2D CNN with 64 latent dimensions followed by a batch norm and leaky-ReLU (parameter=0.1). A 2x2 max-pool and $50\%$ dropout is done after each module and finally a max-pool to get a 64 dimensional latent vector, $z$.

The Decoder is comprised of 3 layers of 3x3 transposed convolutions with dimensionalities: (64, 64, 32) each followed by a batch normalization and ReLU. Finally, a last transposed convolution from 64 to 3 dimensions and a ReLU.

**Training hyperparameters:**  The networks were implemented using the PyTorch framework Paszke et al. (2017). Pre-training on simulated data and refinement on real data were done using the Adam optimizer Kingma & Ba (2014) with default parameters, with learning rates of $1e-3$ and $1e-4$ respectively. Pretraining was done for 400 epochs and refinement was done for roughly 50 epochs. The implementation was done

The classifier and the latent canonicalizers are simply single linear layers with 64 dimensions (corresponding to the dimensionality of the latent, $z$).

## C    SIMULATOR DETAILS

To support our training requirements, we built a generator for SVHN-like images. Importantly, we do not aim to mimic the true SVHN dataset statistics. We make only very general simplistic assumptions such as a 32x32 image size, that images contain a centralized digit, etc. We build the simulator based on the *imgaug* library (Jung et al., 2019) which allows to easy control of the different factors of variation. The specific factors used to generate the images are given in the table together with their set of values. The first row of Table A3 shows the factors used for canonicalization. For each of these (except for noise, blur and crop which are canonicalized together to serve as the bypass), we generate a copy of the image with that factor set to its canonical value. The canonical values are reported in row 3 of the table. They are chosen arbitrarily but are the same for the entire train set. Following the 73257 real SVHN train set samples (not used in this work) we generated a synthetic set of size 75000 total images. Some examples of generated images can be seen in Figure A1. The set together with the simulator code will be made publicly available.

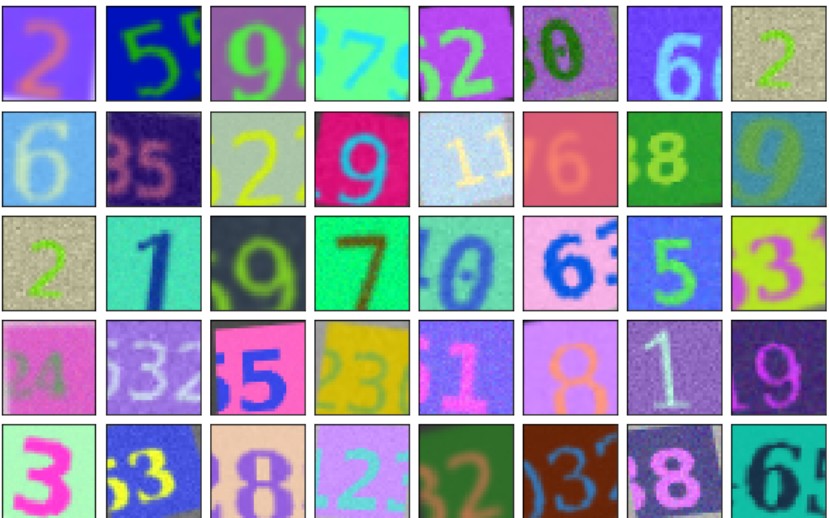

**Figure A1: Examples of simulated SVHN images.**

| Supervised factors | Rotation | Font color | Background color | Font scale | font type | shear | Noise , Blur , Crop |
|---|---|---|---|---|---|---|---|
| Range | $[-15, 15]$ | $\{0, .., 255\}^3$ | $\{0, ..., 255\}^3$ | $[1.0, 1.6]$ | 6 types | $[-5, 5]$ | $\mathcal{N}(0, 0.04)$ , $[0, 1.5]$ , $[0, 0.1]$ |
| Canonical value | 0 | $(200, 200, 200)$ | $(100, 100, 100)$ | 1.0 | type 1 | 0 | $(0, 0, 0)$ |

| Unupervised factors | Cval | Scale | Number of digits | Translation |
|---|---|---|---|---|
| Range | $0, ..255$ | $[0.95, 1.2]$ | $1, ..., 4$ | $\pm Poiss(1)$ |

**Table A3: Simulator factors and their range of values.**

**Domain gap between synthetic and real images:**    Having created quite a generic set of digit images with naive assumptions, one may ask whether it is at all useful for the target domain. By taking intermediate checkpoints during training and measuring the classification error without fine-tuning on the real test set, we see in Figure A2 that indeed there is good correlation the train set error and the test set accuracy.

**Sampling the space of compositions:**    Scene complexity grows exponentially with the number of factors. One goal of this work is to show we can get good performance even when sampling a fraction of this space. To get a rough estimate of that percentage we can bin even just the supervised factors to [30, 64, 64, 6, 6, 10] (following the order of table A3) discrete values. This gives a space of $\approx 44M$ possibilities. Noticing the coarse binning of color-space and the fact we haven't included the unsupervised factors, we conclude that our 75000 samples set size is at least 3 order of magnitude smaller than the number of possible factor combinations.

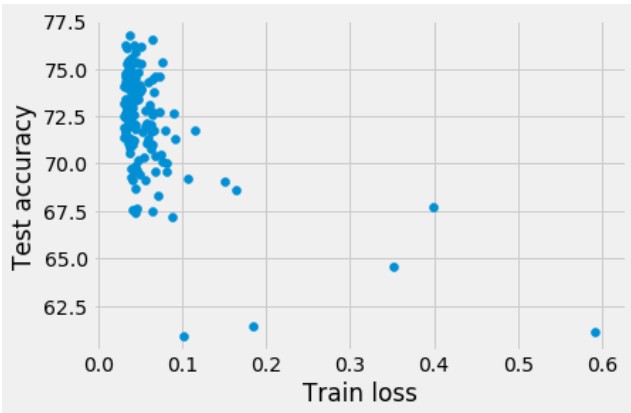

**Figure A2: Train error on synthetic images vs. Test accuracy on real SVHN images.**

## D   ANALYSIS OF REPRESENTATIONS – ADDITIONAL MATERIAL

Bypass

Rotation

Font size

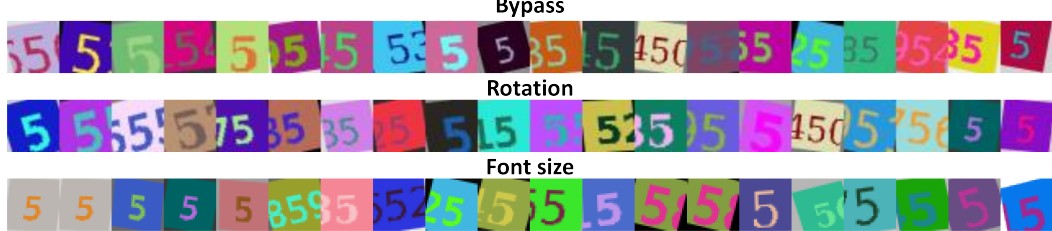

**Figure A3: Comparison between the representation learned by the canonicalizers and the bypassed representation.** Here we show an extended version of Figure 5 also visualizing the bypassed latent code (top row). The first principal component of $z$ does not show any obvious grouping or pattern.

