# OpenReview forum: "Representation Learning Through Latent Canonicalizations"
_ICLR.cc/2020/Conference — Reject_

### Official Review · AnonReviewer2 · 2019-10-22
**Official Blind Review #2**

**Rating:** 3

**Review:**


1. I had hard time to understand latent canonicalization. Do you mean that each latent variable is fixed to a value, such that this factor of variation is disabled? Are the canonicalizers pre-specified using meta-labels? Are they updated/learned during model training?   More explanation of canonicalization is needed. Perhaps an example in linear algebra is needed.

2. The learning of the proposed model relies on meta-data description such that the learning is supervised. Can the method be applicable to situations where no meta-data and no class labels are available?

3. How can the proposed method be generalized to non-image data? The experiments were only done on simple image datasets. I am wondering this method can be applied to other complex datasets whose latent factors are unknown.

4. I do not understand this: "to fit well the method overfitting rate" in Section 3.3.

Minors:
(1) than -> that
(2) Eq. (3): is there a superscription "(j)" on z_canon in decoder?

**Experience Assessment:**

I have published one or two papers in this area.

**Review Assessment: Checking Correctness Of Derivations And Theory:**

I carefully checked the derivations and theory.

**Review Assessment: Checking Correctness Of Experiments:**

I carefully checked the experiments.

**Review Assessment: Thoroughness In Paper Reading:**

I read the paper at least twice and used my best judgement in assessing the paper.

---

> ### Author Response · Authors · 2019-11-09
> **Response to R2**
>
> Q: "I had hard time to understand latent canonicalization..."
>
> A: "latent canonicalization" is a procedure by which the representation is being modified such that a factor of variation assumes a certain prescribed value. While the values of the factors are pre-specified, the canonicalizers are learned linear operators.
>
> Q: The learning of the proposed model relies on meta-data description such that the learning is supervised. Can the method be applicable to situations where no meta-data and no class labels are available?
>
> A: as we discuss in the general comments, the method is best suited for problems of interest where a synthetic simulator is available. In those cases full access to factors of variation is available as this is used to generate the data., The question we focus on is how to best utilize the knowledge of factors of variation to improve the transferability of the learned representation to real data with limited labels. We are not focusing on a setting where the source domain has no labels. That said, our method does not assume access to all factors for all samples (in fact we only use two factors per sample).
>
> Q: How can the proposed method be generalized to non-image data? The experiments were only done on simple image datasets. I am wondering this method can be applied to other complex datasets whose latent factors are unknown.
>
> A: Our method is not specific to the image domain and there is no reason it could not be applied to other input types. In order to use our method, we only require a corpus of data with partially (at least two) labeled factors of variation and a corpus of data which is sparsely labeled for a downstream task. Critically, these two sources of data need not be identical! This means that an easy way to acquire a corpus with labeled factors of variation is to use a simulator to generate this data,  as we do in this study.
>
>
> Q: I do not understand this: "to fit well the method overfitting rate" in Section 3.3.
>
> Thank you for pointing this sentence out! It is indeed unclear. All we were trying to say is that each baseline’s training duration was chosen independently to prevent overfitting. We have updated the draft to be more clear.

---

### Official Review · AnonReviewer1 · 2019-10-25
**Official Blind Review #1**

**Rating:** 8

**Review:**

This paper proposes a method for unsupervised learning  of data representations that can be manipulated to remove factors of variation via linear transformations. These transformations are called canonicalizations in the paper. The canonicalizations are trained such that images for arbitrary values of the corresponding factor of variation are transformed into images with a fixed, canonical, value for that factor. The paper proposes a model architecture based on a denoising autoencoder, where the canonicalizations are applied to the encoded representation. It also proposes a loss function and sampling scheme for training the model. The paper demonstrates the method on the dSprites dataset, showing that it can effectively learn linear canonicalizations, and that multiple of these canonicalizations can be applied to the same image representation. The paper goes on to test the method on a digit classification task, where the model is trained to learn a representation in a simulator for SVHN data where the transformations to be canonicalized can be controlled, and  used to train a classifier on unseen real data from the SVHN test set.

I think this paper should be accepted as it proposes a novel idea, which does not seem too difficult to reproduce, describes a simulator for synthetic data for digit recognition, and proposes it as a benchmark for learning representations, and provides experimental results that help in better understanding the representation learned by the model.


A couple things I thought were missing in the paper:

Did you try applying the classification loss to both the encoded representation and the canonicalized representation at the same time?

For Figure 2, Did you try applying canonicalizations in different orders? Do they give the same results?

Instead of trying to learn idempotency by gradient descent, you could try to parametrize the canonicalizations with a matrix X, such that C =  X (X^T X)^{-1} X^T. C will be idempotent (although restricted to be symmetric). There might be other constructions that are more efficient and less restrictive.

I'm not sure I understand the PCA figures. Can you please explain how the first principal component was used to generate them?


Minor comments:

* "data  tripets" on page 2
* Figure 5 should appear after Figure 4.

**Experience Assessment:**

I have read many papers in this area.

**Review Assessment: Checking Correctness Of Derivations And Theory:**

N/A

**Review Assessment: Checking Correctness Of Experiments:**

I carefully checked the experiments.

**Review Assessment: Thoroughness In Paper Reading:**

I read the paper thoroughly.

---

> ### Author Response · Authors · 2019-11-09
> **Response to R1**
>
> Q: Did you try applying the classification loss to both the encoded representation and the canonicalized representation at the same time?
>
> A: Yes we did and interestingly, it didn't show improvement. The results are reported in Appendix A, Table A1 in the row titled: "Ours + classifier after" and discussed in Section 4.2.4. Note that because the bypassed latent, z, is included along with the canonicalized latents, z_canon, the classifier is trained on both the original representation and the canonicalized representations together.
>
> Q: For Figure 2, Did you try applying canonicalizations in different orders? Do they give the same results?
>
> A: For our experiments, each training example was bypassed and canonicalized by four different transformations: C_1, C_2, C_1C_2, and C_2C_1. So latents are canonicalized in both possible orderings. We discuss this point at the bottom of page 4 after equation 6 and will further clarify.
>
> Q: Instead of trying to learn idempotency by gradient descent, you could try to parametrize the canonicalizations with a matrix X, such that C =  X (X^T X)^{-1} X^T. C will be idempotent (although restricted to be symmetric). There might be other constructions that are more efficient and less restrictive.
>
> A: Structuring the matrix to enforce properties like idempotency is definitely an interesting direction to explore in future work. While not equivalent, as a first step towards this, we did try to examine different levels of idempotency enforcement. First, at the reconstruction loss, since pairs of canonicalizers are applied in different order we enforce the the decoded results are similar. However, adding a stricter enforcement of similar values before decoding only hurt performance, hinting that the suggested idea of enforcing idempotency at the strictest level may hurt performance further.
>
> Q: I'm not sure I understand the PCA figures. Can you please explain how the first principal component was used to generate them?
>
> A: Treating a canonicalizer as a standard linear projection, we explore z - P*z which should contain only the factor of interest (intuitively it is the difference between a representation and a version of it stripped off of the specific factor so that the factor is isolated). Creating a set of such latent samples (here we took all examples to be of the same digit for visualization purposes) we ran PCA to get a dimension with the most significant variance. If the above mentioned assumption indeed holds, the font should be the only change in the set. We order the samples according to that axis and plot them in a row from left to right. We see that indeed the font (bottom row) and angle (top row) present a good correlation with the value along the axis.

---

### Official Review · AnonReviewer4 · 2019-10-28
**Official Blind Review #4**

**Rating:** 3

**Review:**

This paper proposes to relax the assumption of disentangled representation and encourage the model to learn linearly manipulable representations.
The paper assumes that the latent canonicalizers are predefined for each task and that it is possible to obtain the ground-truth image of different latent canonicalizations. I find these assumptions too strong for the task of learning disentangled representation.
Firstly, most prior works such as beta-vae, info-gan do not assume that the factors / canonicalizers are known beforehand. In fact, this is a very difficult part of learning disentangled representation. Secondly, if it is possible to obtain the ground-truth image of different latent canonicalizations, you can simply train a network to predict the canonicalizations by simple supervised learning. Hence, these overly simplified and unrealistic assumptions make the task too trivial.
The proposed method is very simple and frames the problem basically as a supervised learning problem. Although experiments show that learning such representations are beneficial for low-shot setting of SVHN, it is not clear whether such improvement generalizes to more realistic datasets such as ImageNet. If the goal is to learn representation for low-shot setting, the method needs to be compared with other representation learning methods such as jigsaw[1], colorization[2] and rotation[3].

[1] Noroozi, Mehdi, and Paolo Favaro. "Unsupervised learning of visual representations by solving jigsaw puzzles." European Conference on Computer Vision. Springer, Cham, 2016.
[2] Zhang, Richard, Phillip Isola, and Alexei A. Efros. "Colorful image colorization." European conference on computer vision. Springer, Cham, 2016.
[3] Gidaris, Spyros, Praveer Singh, and Nikos Komodakis. "Unsupervised representation learning by predicting image rotations." arXiv preprint arXiv:1803.07728 (2018).

**Experience Assessment:**

I have read many papers in this area.

**Review Assessment: Checking Correctness Of Derivations And Theory:**

I assessed the sensibility of the derivations and theory.

**Review Assessment: Checking Correctness Of Experiments:**

I carefully checked the experiments.

**Review Assessment: Thoroughness In Paper Reading:**

I read the paper at least twice and used my best judgement in assessing the paper.

---

> ### Author Response · Authors · 2019-11-09
> **Response to R4**
>
> Q: "...I find these assumptions too strong for the task of learning disentangled representation."
>
> A: We wish to emphasize that:
> (1) we only require access to meta-labels on the source set
> (2) our goal is not to find disentangled representations; Our goal is transferability so that we can learn on real data with minimal supervision.
>
>
> Q: "you can simply train a network to predict the canonicalizations by simple supervised learning"
>
> A: Finding the canonicaliers is not our goal. These are used as auxiliary constraints which guide representation learning and allow for latent data augmentation on our limited target data. Specifically, predicting the factor values will not help us in manipulating the target samples (note the significant improvement we get by using the majority vote).
>
> Q: method needs to be compared with other representation learning methods such as jigsaw[1], colorization[2] and rotation[3].
>
> A: The suggested methods [1-3] are self-supervised methods using less information than the baselines we have included (for example, the AE+classifier baseline is trained on the same synthetic data with the same access to digit labels). We therefore expect that these methods will perform worse than our baselines. We have included  a comparison with method [3] (see general comment to all reviewers).

---

### Author Response · Authors · 2019-11-09
**General response to all reviewers**

We would like to thank the reviewers for their comments and valuable suggestions. Before answering each of the individual reviewers, we wish to make a general clarification of our main contribution:

As nicely summarized by R1, this paper explores a new method for learning semantically meaningful representations. There are several fundamental differences between our approach and the traditional unsupervised learning of disentangled representations.

First, we explore the notion of constraining the way a representation can be manipulated rather than constraining the structure of the representation itself by introducing “latent canonicalizers”. These are learned linear transformations that operate on the latent representation to manipulate single factors of variation independently, allowing us to relax the usual assumptions of explicit axis-alignment latent disentanglement and instead find a linear subspace per-factor where that factor is canonicalized.

Second, in order to learn these latent canonicalizers, we assume access to the meta-labels on a source set. Critically, for the target set we only assume the task labels (e.g. class). We then use this for supervision, to answer the following question: How should one best utilize labeled factors of variation on a source set, to achieve good performance on a limited labelled target set? Usually the source set would be simulated, in which case we can in fact control all those factors, but other densely labelled sources could be used as well, like facial attributes. Importantly, we accommodate unknown factors by using the bypass channel. This is also demonstrated in our experiments where we only supervise for 6 out of 12 factors. Simulators are widely used to provide additional supervised data for solving tasks which lack real-world annotations.   Most prior works use only a single factor, that which relates to the end task,  as simulated supervision. But, at the same time, one has access to a set of factors to vary in order to generate a diverse set of simulated images. Here we propose an approach which effectively uses the other known factors of variation to impose constraints on the downstream task learning such that the resulting representations are more transferrable and lead to better downstream performance on a sparsely labeled target data source.

Third, we focus our attention to a downstream task. As shown recently, having good disentanglement measured by common metrics does not necessarily mean good performance on downstream task [1]. Instead, here we demonstrate a simple technique with significant improvement over baselines, for transferability between simulated and real data.

[1] Challenging Common Assumptions in the Unsupervised Learning of Disentangled Representations

---

> ### Author Response · Authors · 2019-11-15
> **results on requested experiment**
>
> Following the suggestion from R#4, we have trained an unsupervised method based on "Unsupervised representation learning by predicting image rotations." using our simulated data. As with the other baselines, we evaluated this pre-training stage on a few-shot setting with real SVHN, by using the learned representation to initialize a classification network. Evaluated on 20, and 100 shot respectively this method achieves 66.42 (std: 2.16) and 86.26 (std: 1.11) respectively. In both cases the gap from our approach is significant (17 and 3.5 pts). Interestingly though, while on 20 shot this representation performed worse than an AE + classifier it did better than a vanilla AE. On the 100 shot, it already did better than the other baseline by a small margin.
>
> We appreciate the idea for this experiment and we will include it in the main manuscript.

---

### Decision · Program_Chairs · 2019-12-19

**Decision:**

Reject

**Comment:**

This paper proposes a method to allow models to generalize more effectively through the use of latent linear transforms.

Overall, I think this method is interesting, but both R2 and R4 were concerned with the experimental evaluation being too simplistic, and the method not being applicable to areas where a good simulator is not available. This seems like a very valid concern to me, and given the high bar for acceptance to ICLR, I would suggest that the paper is not accepted at this time. I would encourage the authors to continue with follow-up experiments that better showcase the generality of the method, and re-submit a more polished draft to a conference in the near future.